# In Vitro and In Silico Screening of 2,4,5-Trisubstituted Imidazole Derivatives as Potential Xanthine Oxidase and Acetylcholinesterase Inhibitors, Antioxidant, and Antiproliferative Agents

**Eduardo Noriega-Iribe** [1] **, Laura Díaz-Rubio** [1,*] **, Arturo Estolano-Cobián** [1] **,**
**Victor Wagner Barajas-Carrillo** [1] **, José M. Padrón** [2] **, Ricardo Salazar-Aranda** [3] **,**
**Raúl Díaz-Molina** [4] **, Victor García-González** [4] **, Rocio Alejandra Chávez-Santoscoy** [1,5] **,**
**Daniel Chávez** [6] **and Iván Córdova-Guerrero** [1,*]

[1] Facultad de Ciencias Químicas e Ingeniería, Universidad Autónoma de Baja California, Tijuana 22390, Mexico; eduardo.noriega@uabc.edu.mx (E.N.-I.); arturo.estolano@uabc.edu.mx (A.E.-C.); wbarajas@uabc.edu.mx (V.W.B.-C.); ale.santoscoy@gmail.com (R.A.C.-S.)
[2] BioLab, Instituto Universitario de Bio-Orgánica "Antonio González" (IUBO-AG), Universidad de La Laguna, c/Astrofísico Francisco Sánchez 2, 38206 La Laguna, Spain; jmpadron@ull.es
[3] Departamento de Química Analítica, Facultad de Medicina, Universidad Autónoma de Nuevo León, San Nicolás de los Garza, Monterrey 64460, Mexico; salazar121212@yahoo.com.mx
[4] Departamento de Bioquímica, Facultad de Medicina Mexicali, Universidad Autónoma de Baja California, Mexicali 21000, Mexico; rauldiaz@uabc.edu.mx (R.D.-M.); vgarcia62@uabc.edu.mx (V.G.-G.)
[5] Tecnológico de Monterrey, Escuela de Ingeniería y Ciencias, Av. Eugenio Garza Sada 2501 Sur, Monterrey 64849, Mexico
[6] Centro de Graduados e Investigación en Química, Tecnológico Nacional de México/Instituto Tecnológico de Tijuana, Tijuana 22510, Mexico; dchavez@tectijuana.mx
**\*** Correspondence: ldiaz26@uabc.edu.mx (L.D.-R.); icordova@uabc.edu.mx (I.C.-G.); Tel.: +52-664-120-7741 (I.C.-G.)

**Abstract:** The employment of privileged scaffolds in medicinal chemistry supplies scientists with a solid start in the search for new and improved therapeutic molecules. One of these scaffolds is the imidazole ring, from which several derivatives have shown a wide array of biological activities. A series of 2,4,5-triphenyl imidazole derivatives were synthesized, characterized, and evaluated in vitro as antioxidant molecules using 1,1-diphenyl-2-picrylhydrazyl (DPPH·) and 2-2'-azino-bis-(3-ethylbenzothiazoline-6-sulfonate) (ABTS·+) assays, acetylcholinesterase (AChE) and xanthine oxidase (XO) inhibitors as well as antiproliferative agents. Additional in silico studies such as docking and determination of their absorption, distribution, metabolism, and excretion (ADME) properties were calculated. Compounds **3** and **10** were the most active antioxidants in both the DPPH and ABTS assays (EC$_{50}$ of 0.141 and 0.174 mg/mL, and 0.168 and 0.162 mg/mL, respectively). In the enzymatic inhibition, compound **1** showed the best activity, inhibiting 25.8% of AChE at a concentration of 150 μg/mL, and compound **3** was the most active XO inhibitor with an IC$_{50}$ of 85.8 μg/mL. Overall, against the six different evaluated cancerous cell lines, molecules **2**, **10**, and **11** were the most antiproliferative compounds. In silico predictions through docking point out **11**, and ADME analysis to **11** and **12**, as good candidates for being lead compounds for further derivations.

**Keywords:** imidazole; antiproliferative; antioxidant activities; docking; DPPH; ABTS; acetylcholinesterase; xanthine oxidase

## 1. Introduction

Imidazole (1,3-diaza-2,4-cyclopentadiene) is a heterocyclic aromatic compound that can be found in many biological molecules such as histidine, histamine, or in natural nucleotides. It is a highly versatile pharmacophore; therefore, there are several reports of a wide range of biological activities in molecules containing an imidazole motif such as antifungal, antituberculosis, antibiotic, cytotoxic, anti-inflammatory, antioxidant, and analgesic, amongst many others [1–4].

Imidazole derivatives, being di-, tri-, and tetra-substituted, have shown antioxidant activity through different antioxidant methodologies [5–7]. This is a useful property to counteract oxidative stress, a condition when reactive oxygen species (ROS) overcome the natural cellular antioxidant defense system. As the aging process, along with several chronic and degenerative human diseases, have been linked to oxidative stress such as cardiovascular, neurodegenerative, and cancerous ones [8,9], compounds with antioxidant properties are of high interest for researchers.

One of the neurodegenerative diseases in which oxidative stress has been regarded as one of the underlying causes is Alzheimer's disease (AD) [10], being that this disease is the most frequent cause of dementia in elderly people [11]. As the cholinergic deficit is heavily related to the disease progression, inhibitors of the enzyme acetylcholinesterase (AChEI) are potential drugs for the treatment of AD patients [12]. Imidazole bearing molecules have been also evaluated as AChEI with interesting results [13].

Xanthine oxidase (XO) is a key enzyme in purine metabolism, and is involved in uric acid production as the final metabolite. High production of uric acid can lead to gout; therefore, inhibition of this enzyme has been targeted as a therapeutic approach, with imidazole having been employed for a long time as a scaffold for XO inhibitors [14]. As the activity of XO produces both uric acid and reactive oxygen species, a XO inhibitor with antioxidant properties could show a good therapeutic profile, inhibiting the enzyme and controlling the oxidative damage to tissues near it [14,15].

The literature has shown numerous imidazole derivatives with tri-substitutions, of both alkyl and aryl types, with the aryl types frequently heterocyclic in nature. In a broad sense, in recent years, trisubstituted imidazoles have been synthesized many times, providing new synthetic methodologic alternatives, or in the search of particular biological properties [4]. Alternatively, this article proposes a group of trisubstitutions, where only small variations are introduced in one of them, to conduct a more finely-detailed structure–activity relationship (SAR) of the biological assays performed.

Based on the broad literature for the biological activities of imidazole derivatives and the above-mentioned SAR strategy, in this work, we present the synthesis of 2,4,5-triphenylimidazoles with substitutions in their A ring to perform an initial screening of their activities as antiproliferative, antioxidant, AChE, and XO inhibitor compounds, in order to find new leaders with these biological profiles. To complement the in vitro evaluations, molecular docking and in silico analysis of their ADME properties was made to select the best candidates and set the path for studies on new drug families.

## 2. Materials and Methods

### 2.1. General Information

All reagents for the synthetic methodology and solvents went through purification before being used. Melting point measurements were made on a SMP11 melting point apparatus (Stuart). Different models of UV–Vis spectrophotometers were employed for the UV–Vis spectra, a Genesys 20 model was used for the antioxidant assays, a Microplate reader Multiskan™ FC was used for the acetylcholinesterase assay (both from Thermo Scientific), and a Microplate reader model PowerWave™ XS (from BioTek) was used for the antiproliferative assay and expressed in nanometers (nm). Fourier-transform infrared spectroscopy (FTIR) was performed on a Spectrum One (Perkin-Elmer) and a Nicolet is iS5 spectrophotometer (from Thermo Scientific). Nuclear magnetic resonance (NMR) spectra were obtained on a Bruker spectrometer; model Avance DPX of 400 MHz. The chemical shifts (represented by $\delta$) are shown using tetramethylsilane (TMS) with $\delta$: 0.00 as the internal standard. Gas chromatography mass spectrometry (GCMS) results were obtained on a TRACE 1310

and an ISQ LT models (GC and MS, respectively) from Thermo Scientific. The purification of the synthesized molecules was realized through column chromatography, employing Sigma-Aldrich Silica Gel 60 Å (230–400 mesh). To confirm the achieved purity, compounds were verified by thin-layer chromatography (TLC) employing silica plates backed on aluminum (from Merck), revealing the plates using an UV light at 254 nm.

## 2.2. Synthesis of Triphenyl Imidazole Derivatives

A mixture of ammonium acetate (5 Eq) and acetic acid (10 mL) were refluxed; after five minutes of constant dripping, 1 Eq of the appropriate aldehyde (**1–13**) was added; finally after another five minutes, 1 Eq of benzil was added. Reflux was continued until completion of the reaction (verified by Thin-layer chromatography). To stop the reaction, ammonium hydroxide was added up to a pH of 9, the formed precipitate was filtered, washed using cold water, and dried. To purify the product, column chromatography or recrystallization was employed. Confirmation of all structures were achieved by mass and NMR spectra, as discussed below:

2,4,5-triphenylimidazole (**1**): White powder (yield 95%). $C_{21}H_{16}N_2$. Mp = >250 °C. IR (KBr, $cm^{-1}$) = 3037(C–H aromatic), 1599 (C–C), 1323 (C–N) $cm^{-1}$. $^1$H NMR (400 MHz, CDCl$_3$) δ: 7.97 (d, $J$ = 7.6 Hz, 2H), 7.51 (d, $J$ = 7.2 Hz, 4H), 7.47 (t, $J$ = 7.2 Hz, 2H), 7.40–7.27 (m, 7H). $^{13}$C NMR (100 MHz, CDCl$_3$) δ: 146.29, 132.63, 131.93, 129.04, 128.75, 128.56, 128.21, 127.92, 127.27. GC-MS ($m/z$) = 296 [M]$^+$ (97), 281 (18), 207 (63), 165 (100), 147 (20), 73 (46).

2-(4,5-diphenyl-1*H*-imidazol-2-yl) phenol (**2**): White powder (yield 99%). $C_{21}H_{16}N_2O$. Mp = 210–212 °C. IR (KBr, $cm^{-1}$) = 3205 (O–H), 1601 (C–C aromatic), 1326 (C–N) $cm^{-1}$. $^1$H NMR (400 MHz, CDCl$_3$) δ: 7.70 (dd, $J_1$ = 6.4, $J_2$ = 1.0 Hz, 1H), 7.54 (dd, $J_1$ = 8.0, $J_2$ = 1.6 Hz, 4H), 7.37 (m, 6H), 7.25 (m, 1H), 7.11 (d, $J$ = 8.0 Hz, 1H), 6.86 (dd, $J_1$ = 8.0, $J_2$ = 1.0 Hz, 1H). $^{13}$C NMR (100 MHz, CDCl$_3$) δ: 157.03, 145.17, 130.90, 128.81, 128.38, 128.23, 127.88, 124.30, 119.13, 117.87, 111.81. GC-MS ($m/z$) = 312.3 [M]$^+$ (100), 283.1 (8), 209.1 (4), 165.2 (65), 77.2 (15).

4-(4,5-diphenyl-1*H*-imidazol-2-yl) phenol (**3**): White powder (yield 94%). $C_{21}H_{16}N_2O$. Mp = 248–250 °C. IR (KBr, $cm^{-1}$) = 3423 (O–H), 3056 (C–H aromatic), 1609 (C–C aromatic), 1280 (C–N) $cm^{-1}$. $^1$H NMR (400 MHz, CDCl$_3$) δ: 6.80 (d, 2H), 7.85 (d, $J$ = 8.0 Hz, 2H), 7.46–7.19 (m, 10H). $^{13}$C NMR (100 MHz, CDCl$_3$) δ: 157.27, 146.25, 127.64, 126.56, 121.36, 114.98. GC-MS ($m/z$) = 312 [M]$^+$ (93), 281 (14), 207 (55), 165 (100), 73 (39).

2-(4-methoxyphenyl)-4,5-diphenyl-1*H*-imidazole (**4**): White powder (yield 91%). $C_{22}H_{18}N_2O$. Mp = 230–232 °C. IR (KBr, $cm^{-1}$) = 2958 (C–H aromatic), 1492 (C–C aromatic), 1251 (C–O), 1027 (C–N) $cm^{-1}$. $^1$H NMR (400 MHz, CDCl$_3$) δ: 7.86 (d, $J$= 8.8 Hz, 2H), 7.50 (d, $J$ = 7.2 Hz, 4H), 7.32–7.23 (m, 6H), 6.94 (d, $J$ = 8.8 Hz, 2H), 3.80 (s, 3H). $^{13}$C NMR (100 MHz, CDCl$_3$) δ: 159.83, 146.56, 128.23, 127.89, 127.00, 122.70, 113.97, 55.11. GC-MS ($m/z$) = 326.2 [M]$^+$ (100), 311.2 (32), 283.1 (12), 165.2 (24), 77.1 (9).

2-(3-methoxyphenyl)-4,5-diphenyl-1*H*-imidazole (**5**): White powder (yield 90%). $C_{22}H_{18}N_2O$. Mp = >250 °C. IR (KBr, $cm^{-1}$) = 2998 (C–H aromatic), 2961 (C–H, aliphatic), 1485 (C–C aromatic), 1243 (C–O), 1201 (C-N) $cm^{-1}$. $^1$H NMR (400 MHz, CDCl$_3$) δ: 12.68 (s, 1H), 7.69 (m, 2H), 7.56 (d, $J$ = 7.6 Hz, 2H), 7.51 (d, $J$ = 7.2 Hz, 2H), 7.46 (t, $J$ = 7.2 Hz, 2H), 7.39 (t, $J$ = 7.6 Hz, 2H), 7.31 (t, $J$ = 7.2 Hz, 2H), 7.23 (t, $J$ = 6.8 Hz, 1H), 6.95 (d, $J$ = 7.2 Hz, 1H), 3.83 (s, 3H). $^{13}$C NMR (100 MHz, CDCl$_3$) δ: 160.02, 145.81, 137.52, 135.60, 132.10, 131.53, 130.25, 129.11, 128.95, 128.63, 128.26, 127.52, 126.96, 118.08, 114.67, 110.64, 55.67. GC-MS ($m/z$) = 326 [M]$^+$ (100), 282 (10), 207 (10), 165 (74), 89 (28), 77 (21), 44 (61).

2-(2-methoxyphenyl)-4,5-diphenyl-1*H*-imidazole (**6**): Pale yellow powder (yield 93%). $C_{22}H_{18}N_2O$. Mp = 200–202 °C. IR (KBr, $cm^{-1}$) = 1601 (C–C aromatic), 1480 (C–C aromatic), 1240 (C–O), 766 (C–H) $cm^{-1}$. $^1$H NMR (400 MHz, CDCl$_3$) δ: 10.48 (s, 1H), 8.48 (dd, 1H, $J_1$ = 8 Hz, $J_2$ = 1.2 Hz), 7.66–7.49 (m, 4H), 7.33–7.23 (m, 7H), 7.11 (t, $J$ = 7.6 Hz, 1H), 7.00 (d, $J$ = 8.4 Hz, 1H), 3.99 (s, 3H). $^{13}$C NMR (100 MHz, CDCl$_3$) δ: 155.65, 143.98, 129.46, 128.54, 127.72, 121.58, 118.05, 111.12, 55.80. GC-MS ($m/z$) = 326.3 [M]$^+$ (100), 308.2 (80), 295.1 (39), 221.2 (39), 165.2 (57), 77.2 (16).

4-(4,5-diphenyl-1*H*-imidazol-2-yl)-2-methoxyphenol **(7)**: White powder (yield 92%). $C_{22}H_{18}N_2O_2$. Mp = 246–248 °C. IR (KBr, cm$^{-1}$) = 3510 (O–H), 2996 (C–H), 1601 (C–C aromatic), 1496 (C–C aromatic), 1274 (C–O), 695 (C–H) cm$^{-1}$. $^1$H NMR (400 MHz, CDCl$_3$) δ: 7.57 (d, *J* = 1.6 Hz, 1H), 7.49 (d, *J* = 6.8 Hz, 4H), 7.36–7.23 (m, 7H), 6.88 (d, *J* = 8.4 Hz, 1H), 3.88 (s, 3H). $^{13}$C NMR (100 MHz, CDCl$_3$) δ: 147.23, 146.75, 146.63, 132.47, 128.26, 127.91, 127.11, 121.78, 118.60, 114.76, 109.08, 55.71. GC-MS (*m/z*) = 342 [M]$^+$ (4), 341 (15), 311 (14), 295 (5), 165 (30), 105 (100), 77 (56), 44 (29).

2-(3,4-dimethoxyphenyl)-4,5-diphenyl-1*H*-imidazole **(8)**: White powder (yield 84%). $C_{23}H_{20}N_2O_2$. Mp = 220–222 °C. IR (KBr, cm$^{-1}$) = 2959 (C–H), 1591 (C–C aromatic), 1495 (C–C aromatic), 1253 (C–O), 762 (C–H) cm$^{-1}$.$^1$H NMR (400 MHz, CDCl$_3$) δ: 7.60 (s, 1H), 7.49 (m, 5H), 7.32 (m, 6H), 6.88 (d, *J* = 8.4 Hz, 1H), 3.89 (s, 3H), 3.86 (s, 3H). $^{13}$C NMR (100 MHz, CDCl$_3$) δ: 149.28, 148.90, 146.51, 132.66, 128.23, 127.90, 127.05, 122.95, 118.12, 111.00, 109.04, 55.68. GC-MS (*m/z*) = 356 [M]$^+$ (1), 342 (1), 281 (3), 207 (16), 193 (12), 176 (100), 165 (14), 69 (65).

2-(2-chlorophenyl)-4,5-diphenyl-1*H*-imidazole **(9)**: Pale yellow powder (yield 93%). $C_{21}H_{15}ClN_2$. Mp = 190–192 °C. IR (KBr, cm$^{-1}$) = 2924 (C–H), 1602 (C–C aromatic), 1479 (C–C aromatic), 763 (C–H), 696 (C–Cl) cm$^{-1}$. $^1$H NMR (400 MHz, CDCl$_3$) δ: 10.25 (s, 1H), 8.44 (dd, 1H, $J_1$ = 7.6 Hz, $J_2$ = 1.2 Hz), 7.66 (m, 2H), 7.47–7.25 (m, 11H). $^{13}$C NMR (100 MHz, CDCl$_3$) δ: 143.20, 137.91, 134.57, 130.88, 130.48, 129.59, 129.57, 129.04, 129.02, 128.36, 128.06, 127.96, 127.79, 127.67, 127.52, 127.09. GC-MS (*m/z*) = 330 [M]$^+$ (18), 281 (18), 207 (65), 176 (71), 165 (61), 89 (35), 44 (100).

4-(4,5-diphenyl-1*H*-imidazol-2-yl)-N,N-dimethylaniline **(10)**: Brown yellow powder (yield 83%). $C_{23}H_{21}N_3$. Mp = 234–236 °C. IR (KBr, cm$^{-1}$) = 3000 (C–H), 1618 (C–C aromatic), 1497 (C–C aromatic), 1200 (C–N), 765 (C–H), 696 (C–Cl) cm$^{-1}$. $^1$H NMR (400 MHz, CDCl$_3$) δ: 9.70 (s, 1H), 7.77 (d, *J* = 8.8 Hz, 2H), 7.51 (d, *J* = 7.2 Hz, 4H), 7.30–7.21 (m, 6H), 6.70 (d, *J* = 8.8 Hz, 2H), 2.96 (s, 6H). $^{13}$C NMR (100 MHz, CDCl$_3$) δ: 150.74, 147.01, 132.98, 129.94, 129.06, 128.48, 127.85, 127.15, 126.56, 112.14, 40.32. GC-MS (*m/z*) = 339 [M]$^+$ (6), 325 (4), 313 (17), 269 (23), 178 (53), 165 (100), 89 (42), 77 (38).

2-(4-nitrophenyl)-4,5-diphenyl-1*H*-imidazole **(11)**: Yellow powder (yield 90%). $C_{21}H_{15}N_3O_2$. Mp = 230–232 °C. IR (KBr, cm$^{-1}$) = 2923 (C–H), 1600 (C–C aromatic), 1519 (N–O), 1486 (C–C aromatic), 1339 (N–O), 765 (C–H) cm$^{-1}$. $^1$H NMR (400 MHz, CDCl$_3$) δ: 8.19 (d, *J* = 8.8 Hz, 2H), 8.06 (d, *J* = 8.8 Hz, 2H), 7.53–7.45 (m, 4H), 7.33 (m, 6H). $^{13}$C NMR (100 MHz, CDCl$_3$) δ: 147.03, 143.89, 135.83, 128.53, 128.04, 125.70, 124.14. GC-MS (*m/z*) = 341 [M]$^+$ (1), 330 (10), 281 (4), 220 (10), 176 (100), 165 (24), 89 (21), 69 (60), 45 (40).

2-(2-nitrophenyl)-4,5-diphenyl-1*H*-imidazole **(12)**: Red powder (yield 77%). $C_{21}H_{15}N_3O_2$. Mp = 210–212 °C. IR (ATR diamond, cm$^{-1}$) = 2926 (C–H), 1598 (C–C aromatic), 1517 (N–O), 1485 (C–C aromatic), 1331 (N–O), 759 (C–H) cm$^{-1}$. $^1$H NMR (400 MHz, CDCl$_3$) δ: 8.34 (d, *J* = 8.0 Hz, 2H), 8.11 (d, *J* = 8.0 Hz, 2H), 7.69–7.30 (m, 10 H). $^{13}$C NMR (100 MHz, CDCl$_3$) δ: 147.44, 143.40, 139.83, 135.48, 133.99, 130.27, 129.39, 129.10, 128.57, 128.47, 127.97, 127.71, 127.47, 125.46, 124.43. GC-MS (*m/z*) = 341 [M]$^+$ (48), 311 (10), 237 (5), 165 (28), 135 (31), 104 (100), 89 (60), 79 (29).

2-(anthracen-9-yl)-4,5-diphenyl-1*H*-imidazole **(13)**: Pale yellow powder (yield 64%). $C_{29}H_{20}N_2$. Mp = 204–206 °C. IR (ATR diamond, cm$^{-1}$) = 3074–3020 (C–H), 1609 (C–C aromatic), 1447 (C–C aromatic) cm$^{-1}$. $^1$H NMR (400 MHz, CDCl$_3$) δ: 8.47 (s, 1H), 8.00 (d, *J* = 8.0 Hz, 2H), 7.92 (d, *J* = 8.0 Hz, 2H), 7.64–7.27 (m, 14H). $^{13}$C NMR (100 MHz, CDCl$_3$) δ: 143.60, 131.49, 131.10, 128.91, 128.61, 128.42, 127.83, 127.40, 126.55, 125.83, 125.31, 124.58. GC-MS (*m/z*) = 396.2 [M]$^+$ (100), 323.1 (3), 291.1 (4), 203 (10), 165 (16), 105 (8), 77 (4).

*2.3. In Vitro Antioxidant Activity Assay*

2.3.1. 1,1-Diphenyl-2-Picrylhydrazyl (DPPH) Radical-Scavenging Assay

For the determination of the radical-scavenging activity, we used our implementation of the Salazar-Aranda et al. [16] method. A set of serial dilutions in methanol were prepared for each sample. Then, 0.5 mL aliquots of each dilution were mixed with a solution of 1,1-diphenyl-2-picrylhydrazyl (DPPH) in methanol (0.5 mL, 76 µM). The resulting mixtures were kept in the dark at room temperature

for 30 min. The absorbance of each sample was measured at 517 nm ($A_{517}$) and methanol was used as the blank. To calculate the radical-scavenging activity as DPPH decoloration percentage, the formula below was used:

$$DPPH\ (\%) = [1 - (B/A)] \times 100$$

where A represents the absorbance value of the DPPH solution (used as control) and B is the absorbance of the DPPH solution with the sample. Results were expressed as $EC_{50}$, which represents the required concentration to diminish the absorbance of DPPH by 50%. Quercetin was employed as the reference compound.

### 2.3.2. ABTS Radical-Scavenging Assay

For the determination of the ABTS radical cation (ABTS$^+$) scavenging activity, we used our implementation of the Re et al. and Kuskoski et al. [17,18] method. ABTS$^+$ was produced by reacting an ABTS stock solution (7 mM in water) with 2.45 mM potassium persulfate. The resulting mixture was kept at room temperature in the dark for 16–18 h before its use. Methanol was used to dilute the ABTS$^+$ solution (150 μL) to give an absorbance of $0.7 \pm 0.02$ at 754 nm. This value was taken as the initial absorbance (A1). For each sample, aliquots were prepared mixing 980 μL of the ABTS$^+$ methanolic solution with 20 μL of the samples at diverse concentrations. Each mixture was stirred, incubated at room temperature for 7 min, and its absorbance was read (754 nm). This value was considered as the final absorbance (A2). To calculate the radical-scavenging activity as a percentage of ABTS decoloration, the employed formula was:

$$\%\ of\ inhibition = [(A_1 - A_2)/A_1] \times 100$$

All determinations were performed in triplicate. Results were expressed as $EC_{50}$, which represents the required concentration to diminish the absorbance of ABTS by 50%. Quercetin was used as the reference compound.

### 2.4. In Vitro Acetylcholinesterase Inhibitory Assay

The determination of acetylcholinesterase activity was done using our implementation of the methodology reported by Adewusi et al. [19]. Employing a 96-well plate, 75 μL of Trizma-HCl buffer (50 mM, pH 8) was added along with 75 μL of the synthesized compound diluted, obtaining a 150 μg/mL concentration (0.15% for the dimethyl sulfoxide DMSO) at the end. Subsequently, 25 μL of a buffer solution of 15 mM acetylthiocholine chloride (ATCl) was added to each well with 125 μL of a 3 mM buffer solution of Ellman's reagent (DTNB), giving both of them concentrations of 1.5 mM at the end. Employing a microplate reader every 45 s, the absorbance was measured at a wavelength of 405 nm, for three consecutive times. After these lectures, to each well 25 μL of an enzyme buffer solution with a concentration 2 U/mL of acetylcholinesterase was supplied, enriched with 0.1 mg/mL bovine serum albumin, obtaining an enzyme 0.2 U/mL final concentration. Five consecutive lectures were taken every 45 s. Of each plate, six wells served as the control for the acetylcholinesterase 100% activity, having no tested compound on them. Galantamine was used as the positive control. A correction for the substrate's spontaneous hydrolysis was made by subtracting the absorbance from before the addition of the enzyme from the enzyme containing wells. Using the equation:

$$Inhibition\ \% = 1 - (A_{sample}/A_{control}) \times 100$$

we obtained the percentage of acetylcholinesterase inhibition, where the absorbances were the 0 and 225 s differences of the sample evaluated and for the enzyme 100% activity control previously described. All experiments were performed in triplicate.

### 2.5. In Vitro Xanthine Oxidase Inhibitory Assay

The XO inhibition activity was evaluated using our implementation of the protocol reported by Almada-Taylor et al. [20]. To a volume of 0.33 mL of a xanthine 150 mM solution, phosphate buffer 120 mM with a pH of 7.8 was added (0.4 mL) and mixed with 0.25 mL of a solution of the compound to be analyzed. The reaction was started with the addition of a 0.5 U/mL solution of xanthine oxidase enzyme (0.02 mL). This was allowed to incubate for 3 min at 24 °C, followed by absorbance lecture at 295 nm ($A_{295}$) for the measurement of the formation of uric acid. As a reference, allopurinol was employed, and the control was an absorbance lecture without an inhibitor. Employing the formula:

$$\% \text{ of Xanthine Oxidase inhibition} = [1 - (A_S / A_C)] \times 100,$$

the percentage of xanthine oxidase inhibition activity was determined. $A_S$ indicates the initial velocity of reaction of the sample, and $A_C$ indicates that for the control. All determinations were made in duplicate, and repeated at least three times. Using interpolation from a linear regression analysis, the required concentration to diminish the XO activity by 50% ($IC_{50}$) was calculated.

### 2.6. Cell Lines and Culture Conditions

The in vitro antiproliferative activity of the investigated compounds was evaluated against six human solid tumor cell lines: A549 (non-small cell lung), HBL-100 (breast), HeLa (cervix), and SW1573 (non-small cell lung) as drug sensitive lines; and T-47D (breast) and WiDr (colon) as drug resistant lines. These cell lines were a kind gift from Prof. G. J. Peters (VU Medical Center, Amsterdam, The Netherlands). Cells were maintained in 25 $cm^2$ culture flasks in Roswell Park Memorial Institute (RPMI) 1640 media enriched with 5% FCS (Fetal Calf Serum) and 2 mM L-glutamine in a 37 °C, 5% $CO_2$, and 95% humidified air incubator.

### 2.7. In vitro Antiproliferative Assay

Cells were trypsinized, resuspended in medium containing 5% FCS and antibiotics (100 U/mL of penicillin G and 0.1 mg/mL of streptomycin), counted (Moxi Z automated cell counter), and diluted to reach the appropriate cell densities (2500 cells/well for A549, HBL-100, HeLa and SW1573, and 5000 cell/well for T-47D and WiDr) for inoculation onto 96-well plates. Twenty-four hours later, compounds were added at concentrations in the range 0.01–100 μM. Cisplatin and etoposide were used as the positive control and DMSO (0.25% v/v) was used as the negative control. Drug incubation times were 48 h. Then, cells were fixed using 25 μL ice-cold trichloroacetic acid (TCA) solution (50% w/v) for 60 min at 4 °C, after which time the plates were rinsed with water. Next, 25 μL of a sulforhodamine B (SRB) solution (0.4% w/v in 1% acetic acid) was added for 15 min. Unbound SRB was rinsed with 1% acetic acid. The remaining dye was dissolved with 150 μL of Tris solution (10 mM, pH 10.5). The optical density of each well was determined at 530 and 620 nm using a microplate reader. The anti-proliferative activity, expressed as 50% growth inhibition ($GI_{50}$), was calculated according to NCI formulas [21].

### 2.8. Molecular Docking

The molecular models of the synthesized compounds were obtained inserting their SMILES strings in University of California, San Francisco (UCSF) Chimera 1.11.2 [22]. Energy minimization of the structures was done using Chimera default conditions with Molecular Modelling Toolkit (MMTK) and Antechamber parameters [23]. AutoDock Tools 1.5.6 [24] was employed to define the rotatable bonds and atomic charges for each ligand. Download of the crystallographic structures of the receptors EGFR (PDB ID: 4HJO) and HER2 (PDB ID:3PP0) was done through Protein Data Bank (https://www.rcsb.org/) [25]. Each receptor was prepared with AutoDock Tools, removing the co-crystalized ligand along with the molecules of water included in the model, adding hydrogens and calculating the Gasteiger charges. AutoDock 4.2 [26] was employed for the docking analysis by using

a grid box of $72 \times 72 \times 72$ Å with x = 24.5, y = 9, z = −1 as the center coordinates for EGFR and x = 17.5, y = 17.5, z = 27 for HER2, with a grid point spacing of 0.375 Å. A Lamarckian genetic algorithm was used with a population size of 150, maximum number of evaluations $2.5 \times 10^6$, maximum number of generations 27000, rate of gene mutation 0.02, and rate of crossover 0.8, generating 10 docked conformations for each analyzed compound.

### 2.9. In Silico Drug-Likeness Prediction

To determine the pharmacokinetics and physicochemical properties related to drug-likeness of the synthesized compounds, the SwissADME web server was employed [27].

## 3. Results and Discussion

### 3.1. Synthesis of Triphenyl Imidazole Derivatives

The 2,4,5-trisubstituted imidazole derivatives **1**–**13** were prepared from a 1,2-diketone (benzil), ammonium acetate and the corresponding aldehydes, following the known Radziszewski reaction and the methodology proposed by Puratchikody et al. with some modifications (Scheme 1) [28], with reaction yields of 64–99%. All compounds were characterized by IR and mass spectroscopy, $^1$H- and $^{13}$C-NMR. In the $^1$H NMR spectra of compounds **1**–**13**, the corresponding signals for the aromatic protons of the rings of position four and five of the imidazole heterocycle were observed, with typical displacements between 7.19–7.69 ppm. For the aromatic system of position two, all of the protons' expected shifts were observed, as were their coupling constants. In the $^{13}$C-NMR spectra, the carbons that formed the imidazole ring were observed at shifts of 159.83–143.20 ppm for carbon two, while those of position four and five were seen at 128.54–127.64 ppm. NMR spectra of the selected derivatives can be observed in Figures S1–S13 in the Supplementary Materials.

| | R₁ | R₂ | R₃ | R₄ | R₅ |
|---|---|---|---|---|---|
| **1** | –H | –H | –H | –H | –H |
| **2** | –H | –H | –H | –H | –OH |
| **3** | –H | –H | –OH | –H | –H |
| **4** | –H | –H | –OCH₃ | –H | –H |
| **5** | –H | –OCH₃ | –H | –H | –H |
| **6** | –OCH₃ | –H | –H | –H | –H |
| **7** | –H | –OCH₃ | –OH | –H | –H |
| **8** | –H | –OCH₃ | –OCH₃ | –H | –H |
| **9** | –Cl | –H | –H | –H | –H |
| **10** | –H | –H | –N(CH₃)₂ | –H | –H |
| **11** | –H | –H | –NO₂ | –H | –H |
| **12** | –NO₂ | –H | –H | –H | –H |
| **13** | | | 9-anthracene | | |

**Scheme 1.** General reaction scheme for the synthesis of 2,4,5-triphenyl-1*H*-imidazole derivatives.

### 3.2. Antioxidant Activity

Both in the DPPH and ABTS assays, imidazole presented $EC_{50}$ of >15 and >10 mg/mL, respectively (Table 1), which compared to most of the results shown by its derivatives, suggests that the 2,4,5-triphenyl substitution in the imidazole heterocyclic is relevant for the antioxidant activity of these compounds, where the effect of their substitutions on their A ring is further developed below.

The DPPH (2,2-diphenyl-1-picrylhydrazyl) radical scavenging method is widely used to evaluate antioxidant activities in a relatively short period of time compared to other methods. The results of this assay are shown in Table 1, comparing the synthesized products with the standard quercetin, where the most active synthesized imidazole derivatives were **3**, **10**, **7**, and **2** with values of $EC_{50}$ of 0.141, 0.174, 0.341, and 1.389 mg/mL, respectively. These results show that the presence of electron donating groups such as hydroxy and *p*-dimethylamino on an aromatic ring bonded to imidazole are essential in the antioxidant activity. The consulted literature indicates that this could be due to the free pair of electrons in nitrogen or in the oxygen of the hydroxy group, which can react with free radicals, being favored due to their aromatic ring stabilization [29]. The rest of the compounds presented low activity in this assay, mainly because of their lack of acidic hydrogen in the aromatic system of position two (A ring); instead, compounds **4**, **5**, **6** and **8** bear methoxy groups, there is a chlorine atom in **9** ($EC_{50}$ of 5.62 mg/mL), an electron attractor effect of the $NO_2$ group in products **11** and **12**, and an anthracene group in **13**.

It is interesting to point out the difference in antioxidant activity between isomers **2** and **3**, where it is shown from the last one that there was a higher oxidative inhibitory potential in both employed techniques (DPPH and ABTS). It is well known that the antioxidant mechanisms of phenolic compounds are hydrogen atom transfer and single electron transfer, in order to inhibit free radicals, which are the expected mechanisms for the phenolic hydroxyls present in these isomers. These different results could be due to the fact that even though both compounds can transfer their hydrogens because of their high acidity, hydroxyl in **2** is in an *ortho* position, favoring the formation of a hydrogen bond along with a nitrogen of the imidazole nucleus, and forming a 6-membered stable ring. It is referenced that these hydrogen bond interactions can diminish the hydrogen dissociation and therefore the antioxidant ability of these groups [30].

In 2015, Hemalatha et al. [31] evaluated the antioxidant activity with a DPPH assay of compounds **2**, **3**, and **10**, reporting $IC_{50}$ values of 0.003, 0.0037, and 0.0031 mg/mL, respectively, while the $IC_{50}$ values for the same compounds in our analysis were 1.389, 0.141 and 0.174 mg/mL, respectively. Even though there were notorious differences between both results, establishing a direct comparation was complicated due to differences in the methodologies employed for the assay, as in [31], a higher concentration of the DPPH radical was employed, and the incubation times for the reactions were not stated.

In a similar way to the DPPH methodology, the ABTS radical-scavenging assay showed that compounds **10**, **3**, **2**, and **7** with $EC_{50}$ values of 0.162, 0.168, 0.188 and 0.199 mg/mL, respectively, were the most active products, however, compound **13** showed moderate activity, while products **1**, **4**, **5**, **6**, **8**, **11**, and **12** presented low activity, as can be seen in Table 1. With these compounds, once again, it is important to emphasize that the participation of hydroxyl and dimethylamino groups play an important role as free radical scavengers. Several reports have discussed the possible mechanisms involved in $ABTS^+$ quenching, suggesting the mixed hydrogen atom transfer/single electron transfer reaction mechanisms [32], and some groups have these properties of chemical reactivity such as *N,N*-dimethylaniline derivatives, which can generate efficient and stable radicals [33].

### 3.3. Acetylcholinesterase Inhibitory Assay

In this assay, galantamine was more active than the products evaluated. Nevertheless, as an initial screening a structure activity relationship is attempted to obtain valuable information for future research.

Among the synthesized products, compound **1** showed the best activity with 25.8% of inhibition (Figure 1). AChE inhibitors bond with the enzyme in a well-known gorge, which in its bottom presents a Trp residue (Trp84 for *Tetronarce californica* AChE, the enzyme used for the in vitro assay). This residue is of crucial importance for ligand interaction by means of a π–cation interaction [34,35]; however, it can also have purely hydrophobic interactions. In the case of galantamine and donepezil [34,36] this residue presents classical π–π stacking with a galantamine double bond, and with the benzyl ring in donepezil. In a similar way, compound **1** could adopt a similar position against AChE, presenting a π–π interaction with Trp84 through its A ring, which has no substitutions that could affect the π electron cloud in the ring, thus explaining the result shown. Hydrophobic and π–π interactions tend to be the most observed ones between AChE and the scaffolds of different inhibitors [37–39].

The next compounds with high inhibition percentages were compounds **11** and **12**, which presented a nitro functionality in their *p*- and *o*- positions. The nitrogen atom in this group is positively charged; in this manner, these compounds could have π–cation interactions with Trp84, or even with Phe330, which is another residue that commonly has this interaction. This could explain why **11** and **12** followed compound **1** with the best results.

Some tendencies seen in the results when comparing **2** (which has an *o*-OH substitution) against **6** (which presents an *o*-OMe one), we can see that the inhibition activity diminishes; the same pattern was observed with **7** and its methoxy version **8**, although the comparison between **3** and **4** appeared as the exception of this behavior. Compound **9** had only 5.9% inhibition activity; as π–π interactions with AChE are important, the chloride presence in **9** could alter the electron cloud from the A ring, disturbing the π–π interactions that can be made.

**Table 1.** Antioxidant activity ($EC_{50}$) of synthesized compounds **1**–**13**.

| Compounds | Antioxidant Activity ($EC_{50}$, mg/mL) | |
| :---: | :---: | :---: |
| | DPPH | ABTS |
| **1** | 3.25 ± 0.137 | 34.312 ± 0.245 |
| **2** | 1.389 ± 0.631 | 0.188 ± 0.011 |
| **3** | 0.141 ± 0.094 | 0.168 ± 0.046 |
| **4** | 16.74 ± 0.003 | 1.644 ± 0.584 |
| **5** | 16.89 ± 0.636 | 37.223 ± 2.629 |
| **6** | 7.12 ± 1.916 | 15.643 ± 0.324 |
| **7** | 0.341 ± 0.101 | 0.199 ± 0.001 |
| **8** | 12.23 ± 3.042 | 1.964 ± 0.37 |
| **9** | 5.62 ± 1.752 | ND |
| **10** | 0.174 ± 0.041 | 0.162 ± 0.006 |
| **11** | ND | 8.025 ± 0.771 |
| **12** | ND | 42.158 ± 2.697 |
| **13** | 4.00 ± 0.135 | 0.449 ± 0.03 |
| Imidazole | >15 | >10 |
| * Quercetin | 0.052 ± 0.037 | 0.075 ± 0.002 |

* Served as the reference compound. Values are mean ± SD, DPPH n = 2, ABTS n = 3. ND = Non-detected in the evaluated concentrations. $EC_{50}$ = Concentration required to decrease the absorbance by 50%.

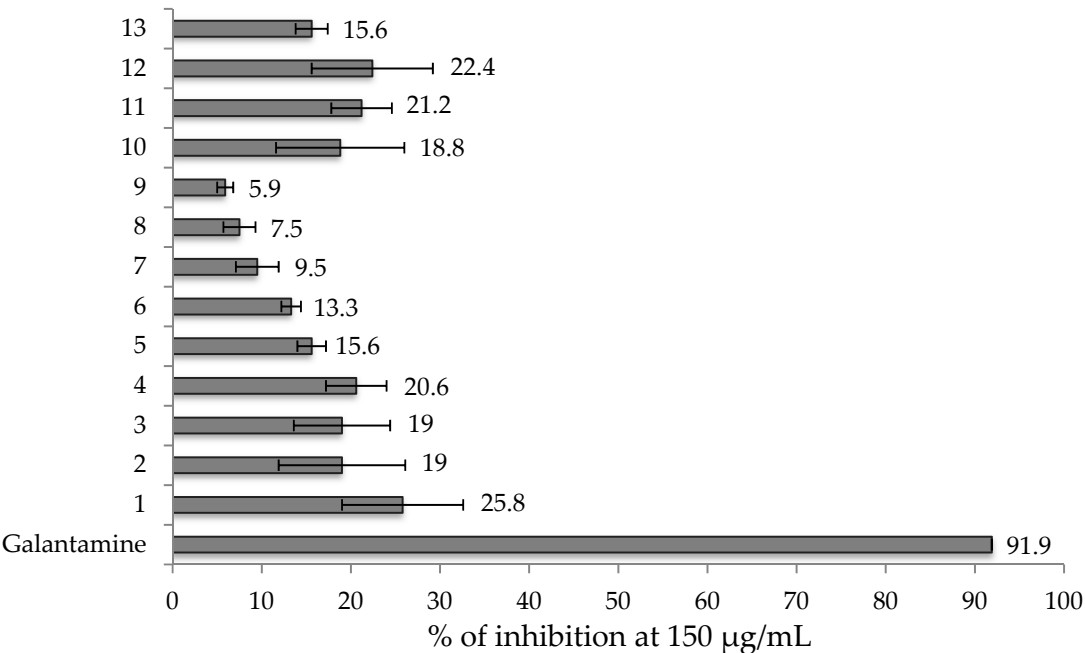

**Figure 1.** Percentage of acetylcholinesterase inhibition of triphenyl imidazole compounds **1–13** (150 µg/mL). Galantamine served as the reference compound.

### 3.4. Xanthine Oxidase Assay

Although not being as active as the positive control allopurinol, some tendencies in the structure activity relationship of the synthesized compounds can be noticed, as seen in Figure 2.

Comparing compounds **2–6** where hydroxy and methoxy substitutions are present, the *p*- substitution can be inferred as a significant requirement for this products, as only *p*-OH and *p*-OMe products showed activity. This was also the case for compounds **7** and **8**, with hydroxy and methoxy groups as substitutions, while having a *para* substitution besides a *meta* one, allowed them to show activity.

It appears that not only the *p*- position is of importance, but also that the functionality in these synthesized compounds must be of -OH or -OMe type, bearing an oxygen as a heteroatom bonded to the aromatic ring. Products **10** and **11** also have substitutions in this position, but with nitrogen as the heteroatom (an amine and nitro group, respectively) and in their case, the *para* position with a nitrogenated group showed no activity. For the synthesized products, the interaction with xanthine oxidase, instead of being similar to the one for allopurinol, which interacts with one of its aromatic nitrogen to bond with molybdenum in the catalytic site of the enzyme [40], could be similar to the topiroxostat one. This inhibitor interacts with the xanthine oxidase molybdenum with its oxygen in a covalent bond [41]. While compound **10** has its nitrogen in a tertiary amine, and **11** in a nitro group, it could be more difficult for them to bond with the Mo center of the enzyme, favoring in our products the presence of oxygen over nitrogen.

Product **3**, having a *p*-OH group and no other substitution that diminishes its activity, resulted in the most active compound from the synthesized ones. Between the hydroxy and methoxy substitutions, it appeared as the first one favored the inhibition activity over xanthine oxidase. Compound **3** with a *p*-OH substitution showed an $IC_{50}$ of 85.8 µg/mL, while **4**, which has a *p*-OMe, showed almost double the $IC_{50}$; again, between **7** and **8**, we could see that the methoxy version was less potent than the hydroxy one. This can be related with the observation made for different products with alcohol groups in their structure such as polyphenols, which can form hydrogen bonds with XO via their hydroxyl groups [14,42].

However, the exception to the structure activity relationship discussed was **12**, having an *o*-NO₂, which lacked a *para* position and oxygen heteroatom functionality. This compound was achieved

as one of the few products with xanthine oxidase inhibition, although it showed the second lowest activity. Further *ortho* nitrogen containing products must be synthesized to expand this analysis.

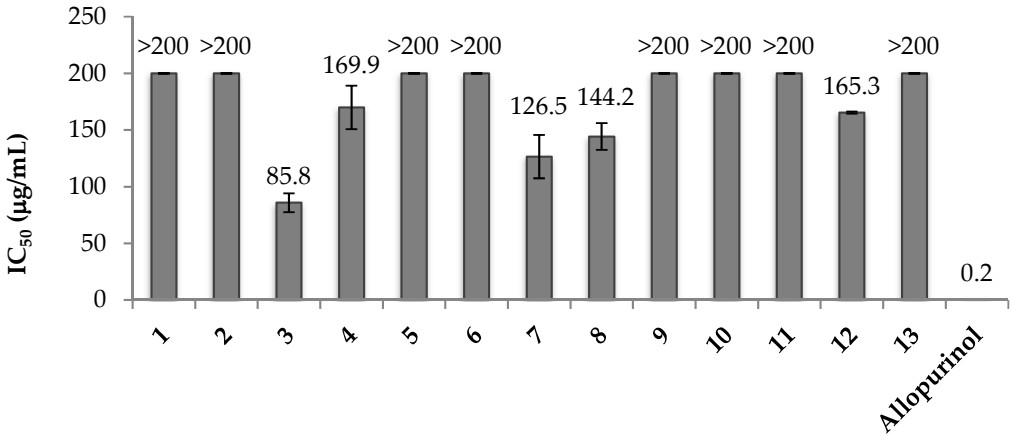

**Figure 2.** Xantine oxidase inhibitory activity of synthesized compounds **1–13**. Allopurinol served as the reference compound. Bars are mean ± SD, n = 3.

### 3.5. Antiproliferative Assay

The antiproliferative activity evaluation of the synthesized triphenyl imidazole derivatives was made with the sulforhodamine B (SRB) assay. The tumoral cell lines employed were adherent epithelial cells from different anatomic origins. All results were expressed as growth inhibition 50 ($GI_{50}$), as the concentration needed to inhibit the 50% of cell population, and calculated and expressed as micromolar ($\mu$M). As positive controls, different antitumor drugs were employed such as cisplatin, etoposide, and camptothecin, and imidazole was used as the structural reference of the synthesized compounds.

The obtained results from the evaluation of the 13 synthesized compounds with the SRB assay against the tumor cell lines (Table 2) showed no selectivity by any specific line. Based on the results in Table 2 and the $GI_{50}$ range (Figure 3), the most active compounds of the series were **10** and **11**. As initial highlights, imidazole had no inhibitory activity in the compounds employed as controls; on the other hand, one of the most resistant cell lines against the synthesized compounds and drugs was A549, which corresponds to lung adenocarcinoma, and this is in agreement with the literature, as it has been documented that lung type cancers are usually chemotherapy resistant, even to one of the most used antitumor drugs, taxol [43].

From the 13 synthesized compounds, lower activity was shown from derivative **1**, this being the triphenyl imidazole bearing no substitutions, as against five of the six evaluated cell lines, it showed no significant activity, and a low one against SW1573 (89 $\mu$M). Likewise, between the molecules with one methoxy substitution **4** (*p*-OMe), **5** (*m*-OMe) and **6** (*o*-OMe), which are position isomers, only **5** showed a low activity against SW1573 with an $GI_{50}$ of 76 $\mu$M, while **4** and **6** presented no significant activity against all of the evaluated cell lines.

Following these general low active compounds, derivatives **9** (*o*-Cl) and **12** (*o*-NO$_2$) were partially active, as they presented different degrees of activity, but against only a couple of cell lines. Compound **9** showed good to moderate activity only against two cell lines, which were HeLa with a $GI_{50}$ of 7.7 $\mu$M and SW1573 with 17 $\mu$M; compound **12** presented activity against the same cell lines with $GI_{50}$ of 6.1 and 66 $\mu$M respectively.

The following molecules with better results were the compounds **3** (*p*-OH), **7** (*m*-OMe, *p*-OH), **8** (*m*-OMe, *p*-OMe), and **13** (anthracene), as these molecules presented activity against all evaluated cell lines, the only exception being **13** against lines T-47D and WiDr; however, **13** also showed one of the best particular results, this being a $GI_{50}$ of 4.2 $\mu$M against SW1573. In the case of **3**, activity was shown against all analyzed cell lines, with HeLa being the most sensitive with a $GI_{50}$ of 13 $\mu$M, followed by

SW1573 and HBL-100 with 15 and 16 μM, respectively, while in the rest of the lines, the results were between 19 and 22 μM.

**Table 2.** Antiproliferative activity of compounds **1–13** against six human solid tumor cell lines [a].

| Compound | Substituent | Cell Lines (GI$_{50}$, μM) | | | | | |
|---|---|---|---|---|---|---|---|
| | | A549 | HBL-100 | HeLa | SW1573 | T-47D | WiDr |
| **1** [b] | -H | >100 | >100 | >100 | 89 | >100 | >100 |
| **2** | *o*-OH | 11 ± 5.5 | **7.0 ± 2.0** | **4.3 ± 0.6** | **3.6 ± 0.2** | 18 ± 0.3 | 19 ± 0.5 |
| **3** | *p*-OH | 19 ± 4.0 | 16 ± 0.4 | 13 ± 2.5 | 15 ± 2.3 | 20 ± 1.5 | 22 ± 0.7 |
| **4** | *p*-OMe | >100 | >100 | >100 | >100 | >100 | >100 |
| **5** [b] | *m*-OMe | >100 | >100 | >100 | 76 | >100 | >100 |
| **6** | *o*-OMe | >100 | >100 | >100 | >100 | >100 | >100 |
| **7** | *m*-OMe, *p*-OH | 17 ± 1.3 | 17 ± 0.7 | 15 ± 1.7 | 15 ± 0.5 | 20 ± 2.0 | 17 ± 1.2 |
| **8** | *m*-OMe, *p*-OMe | 26 ± 1.0 | 15 ± 2.0 | 10 ± 0.4 | 15 ± 3.1 | 16 ± 1.2 | 13 ± 6.8 |
| **9** [b] | *o*-Cl | >100 | >100 | **7.7** | 17 | >100 | >100 |
| **10** | *p*-N(CH$_3$)$_2$ | **3.8** | **5.9 ± 3.1** | **4.5 ± 1.2** | **4.4 ± 2.5** | **5.3 ± 1.9** | **4.5 ± 0.8** |
| **11** | *p*-NO$_2$ | **6.3 ± 3.4** | **3.3 ± 1.6** | **3.0 ± 0.9** | **2.9 ± 0.4** | **5.5 ± 0.2** | **4.6 ± 0.2** |
| **12** [b] | *o*-NO$_2$ | >100 | >100 | **6.1** | 66 | >100 | >100 |
| **13** | 9-anthracene | 45 ± 14 | 23 ± 10 | 12 ± 1.5 | **4.2 ± 1.4** | >100 | >100 |
| Imidazole | - | >100 | >100 | >100 | >100 | >100 | >100 |
| CDDP | - | **4.9 ± 0.3** | **1.9 ± 0.2** | **1.9 ± 0.4** | **2.7 ± 0.4** | 17 ± 2.3 | 23 ± 4.3 |
| VP-16 | - | **1.5 ± 0.3** | **1.2 ± 0.3** | **2.4 ± 0.9** | 15 ± 1.5 | 18 ± 4.4 | 24 ± 2.6 |

[a] GI$_{50}$ values are given in μM. Standard deviation was calculated from two to four independent experiments. Cisplatin (CDDP) and etoposide (VP-16) were used as reference antiproliferative drugs. Values in bold represent the best anti-proliferative data against tumor cell lines with GI$_{50}$ values < 10 μM. [b] Only one experiment was performed.

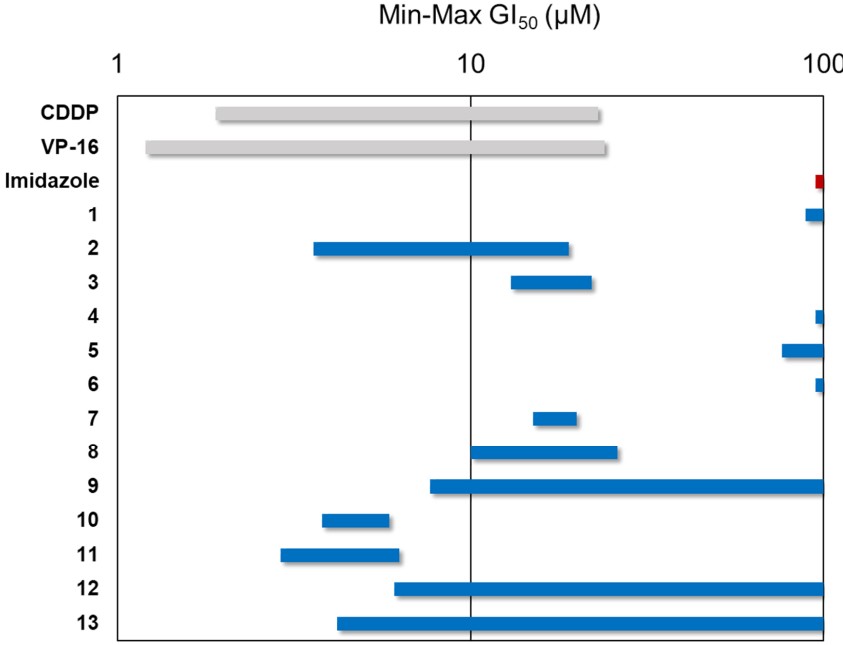

**Figure 3.** GI$_{50}$ range plot of the tested compounds.

With molecules **7** and **8**, very similar GI$_{50}$ values could be seen between them. Comparing them against monosubstituted compounds **4**, **5**, and **6** (which have a methoxy group in different positions), **7** and **8** showed that di-substitution enhanced the antiproliferative activity against these cell lines, these being substitutions of the hydroxy and methoxy type. In the literature, compound **8** was reported to show antiproliferative activity against the breast cancer cell line MDA-MB-231 with a GI$_{50}$ of 21 μM [44]. This value is consistent with the results obtained in our study.

The best set of synthesized molecules were **2** (*o*-OH), **10** (*p*-N(CH$_3$)$_2$) and **11** (*p*-NO$_2$) as they showed the overall lowest GI$_{50}$ values. In particular, **2** presented significant activity against lines SW1573 and HeLa with values of 3.6 and 4.3 μM; **10** showed significant activity against the majority of cell lines, where the outstanding results were against A549 with GI$_{50}$ of 3.8 μM, and against SW1573 with 4.4 μM. The most active compound against all of the analyzed cell lines was **11**, as its GI$_{50}$ was between 2.9 to 6.3 μM, with SW1573 being the most sensitive line.

These two most active compounds had a substitution in their A ring in the *para* position, where the second most active was a tertiary amine nitrogen, and the most active had a nitro group. One of the most sensitive cell lines to the synthesized compounds and the evaluated drugs was SW1573, which is from alveolar carcinoma. This is despite the line belonging to a lung cancer lineage which, as mentioned before with A549, are pharmacotherapy resistant carcinomas. However, these two lung related cell lines showed that they were sensitive to compound **11**, with GI$_{50}$ values of 6.3 for A549 and 2.9 μM for SW1573, which were the second lowest and the lowest values, respectively, for these lines.

In 2017, Dake's research team [45] reported the synthesis and evaluation of triphenyl imidazole derivatives with substitutions in their A ring against the A549 line, where their compound 6f showed an IC$_{50}$ of 15 μM. This molecule has *m*-I, *m*-OMe, and *p*-OH substitutions, where the iodine is structurally similar to **7**. The presence of this heteroatom improved activity by a 2 μM difference compared to not having it (17 μM for molecule **7**).

The *p*-NO$_2$ substitution in compound **11** bears an important role in antiproliferative activity, which could be due to the nitroaromatic structure. Nitroaromatic compounds have gained interest as chemotherapeutic agents against cancer because molecules with nitro groups in their metabolism can go through bio-reduction, which generates reactive species that cause damage to cell components by oxidative stress; additional reductions are favored in hypoxic conditions, which generates highly cytotoxic species [46]. Even though molecule **12** is an isomer of **11**, in comparison, it showed much lower activity. This could be due to **12** having the nitro group in the *ortho* position, where it could interact with the hydrogen in the imidazole ring, diminishing the generation of the reactive species needed for the antiproliferative activity.

### 3.6. Molecular Docking

Encouraged by previous reports from our group where docking techniques were applied with good results [47,48], in the present work, docking was employed to search possible imidazole receptors.

Many solid tumors are characterized by aberrant signal transduction through different receptors belonging to the ErbB family of receptor tyrosine kinases, where the EGFR and HER2 receptors belong; therefore, one therapeutic approach in oncotherapy is the inhibition of one or both of these receptors [49,50]. The ErbB receptors and their ligands are overexpressed in the majority of solid neoplasms; EGFR and ErbB-3 are found on average in 50% to 70% of lung, colon, and breast carcinomas [51]. HER2 is mainly related with breast cancer (is expressed in 30% of primary breast carcinomas [51]), but is also related with ovary, colon, lung, uterine cervix, and esophagus cancers, amongst others [52]. As co-expression of different ErbB receptors occurs commonly, 87% of EGFR positive tumors also express HER2 [51]. Due to all of the above, EGFR and HER2 receptors have been selected in the literature [53] to relate in vitro anti-cancer activity to in silico docking calculations. In this last reference, the results from the docking of imidazole derivatives against EGFR and HER2 showed a general good agreement with their cytotoxic results. They evaluated two imidazoles that are reported in the presented work, **11** and **12**, with generally closely related results; having the same docking algorithm and protocol, differences may arise due to different ligand preparation as this step can influence the final result [54]. In the present work, the proposed docking protocol was employed for an initial screening for both EGFR and HER2 as potential cancer-mediated receptors for the synthesized imidazole derivatives.

The binding energies results from the docking analysis are shown in Table 3, which includes imidazole as a negative control and lapatinib, an EGFR and HER2 inhibitor [55], as the positive control.

All synthesized compounds showed better results than the imidazole, suggesting the derivatization improved their affinity for these receptors. Although lapatinib showed the best result against both enzymes compared to our compounds, it was closely followed by some products. From the synthesized compounds, **11**, **12**, **5**, **9**, and **7** presented the best results interacting with both EGFR and HER2, as they were in the first five places with lower binding energies. After that, there were variations in the order in which the synthesized products interacted with the selected receptors. Comparing the results for the docking in each receptor, against the in vitro results for each of the evaluated cell lines, there was little agreement between them. This can be explained in several ways, one could be the use of a specific docking algorithm, while each one presents differences in the way results are achieved. The employment of different algorithms with the present work dataset of ligand structures and $GI_{50}$ values could be further explored to find the most suitable algorithm for the synthesized ligands. On other hand, it could be possible that the biological receptors where the compounds interact are different to EGFR and HER2, explaining the little correlation shown. Additionally, it has been reported that docking results could be significantly improved with post-docking energy refining through semi-empirical methods such as PM7 [56].

The compound that represented good agreement between its in vitro and in silico results was **11**, bearing a *p*-nitro substitution (Figure 4). It showed −9.11 and −9.19 kcal/mol binding energy with EGFR and HER2, respectively, having the second-best affinity with both receptors. On the other hand, it was the first or second most active compound against the six evaluated cell lines. This suggests that **11** could be one potential lead compound for further derivatization in the search for new active antiproliferative agents.

**Table 3.** Docking scores of the synthesized triphenyl imidazoles with their controls.

| Compound | Binding Energy (kcal/mol) | |
|:---:|:---:|:---:|
| | EGFR | HER2 |
| 1 | −8.32 | −8.59 |
| 2 | −7.92 | −8.92 |
| 3 | −8.24 | −8.1 |
| 4 | −7.89 | −8.4 |
| 5 | −8.87 | −9.13 |
| 6 | −8.2 | −8.58 |
| 7 | −8.68 | −8.99 |
| 8 | −7.63 | −8.99 |
| 9 | −8.49 | −9.1 |
| 10 | −8.28 | −8.98 |
| 11 | −9.11 | −9.19 |
| 12 | −9.88 | −9.31 |
| 13 | −8.37 | −8.23 |
| **Imidazole** | −2.89 | −3.21 |
| **Lapatinib** | −10.48 | −9.88 |

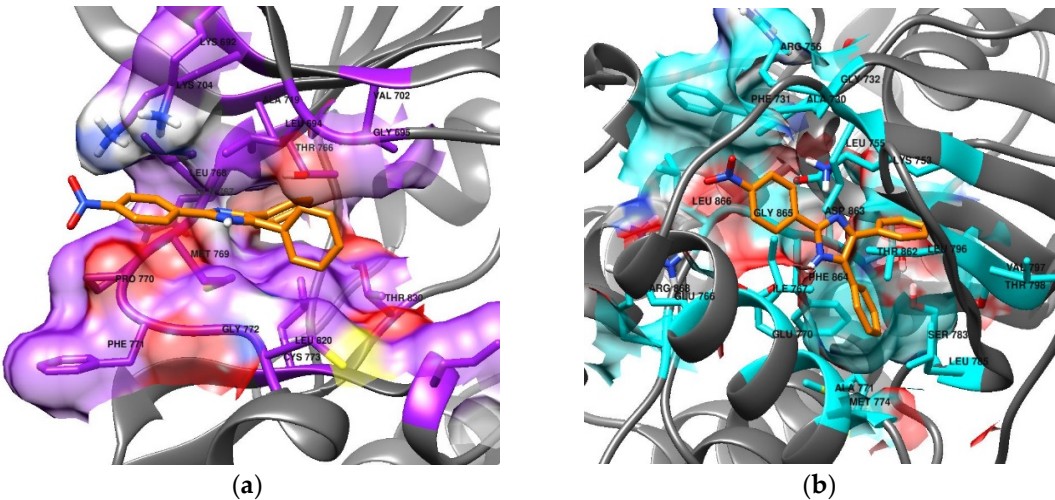

(**a**)  (**b**)

**Figure 4.** Compound **11** (orange) depicted with both of the analyzed receptors. Highlighted are the residues at distances <5.0 Å from **11**, with some surfaces active to show the cavity in which the docking analysis put the best docked pose. (**a**) EGFR, with the near residues in purple. (**b**) HER2, with the near residues in cyan.

## 3.7. In Silico Drug-Likeness Prediction

As can be seen from Table 4, the calculations from the SwissADME website allow for the analysis of which synthesized compounds have better pharmacokinetics and drug-like properties.

All of them had a TPSA between the limits suggested for good bioavailability (20–130 Å$^2$). The vast majority are inhibitors to cytochrome enzymes, which could affect the metabolism and present drug–drug interactions [27], **13** being the least CYP inhibitor, followed by **11**, **12**, and **9**. Although their water solubility was moderate, all of them are predicted to have a high gastrointestinal (GI) absorption (although this can be partially limited for **1**–**10**, being P-gp substrates). The exception to this is compound **13**, which is poorly soluble and has low GI absorption. The great majority seem to be able to permeate the blood–brain barrier (BBB), although this was not the case for compounds **9**, **11**, **12**, and **13**. However, as all the BBB permeant compounds are also P-gp substrates, they would be pumped out from the brain and we would expect no interactions with the central nervous system due to this. Due to these enlisted data, we could expect the synthesized compounds to be, in general, suitable for oral administration.

Lipinski's rule of five [57] can be applied as a first filter, which accounts for the physicochemical properties related to the drug-likeness of a molecule. The molecular weight, number of H-bond donors and acceptors, and lipophilicity are in general accordance to the Lipinski rule. Only compounds **9** and **13** presented a violation, in both cases related to their very high lipophilicity.

Considering the predicted pharmacokinetics and drug-likeness, compounds **11** and **12** with nitro substitution can be considered as promising lead compounds for further studies, which can be additionally supported by the fact they were amongst the most active in vitro compounds, both as AChE inhibitors, **12** as an XO inhibitor, and **11** as part of the antiproliferative imidazoles in cancer cell lines.

**Table 4.** Pharmacokinetics and drug-likeness calculations made by SwissADME for the synthesized compounds **1–13**.

| Descriptors | Compound | | | | | | | | | | | | |
|---|---|---|---|---|---|---|---|---|---|---|---|---|---|
| | **1** | **2** | **3** | **4** | **5** | **6** | **7** | **8** | **9** | **10** | **11** | **12** | **13** |
| MW (g/mol) | 296.37 | 312.36 | 312.36 | 326.39 | 326.39 | 326.39 | 342.39 | 356.42 | 330.81 | 339.43 | 341.36 | 341.36 | 396.48 |
| #H−bond acceptors | 1 | 2 | 2 | 2 | 2 | 2 | 3 | 3 | 1 | 1 | 3 | 3 | 1 |
| #H−bond donors | 1 | 2 | 2 | 1 | 1 | 1 | 2 | 1 | 1 | 1 | 1 | 1 | 1 |
| TPSA ($Å^2$) | 28.68 | 48.91 | 48.91 | 37.91 | 37.91 | 37.91 | 58.14 | 47.14 | 28.68 | 31.92 | 74.5 | 74.5 | 28.68 |
| Consensus Log P | 4.59 | 4.26 | 4.17 | 4.56 | 4.55 | 4.52 | 4.27 | 4.54 | 5.15 | 4.6 | 4 | 3.95 | 6.36 |
| ESOL Log S | −5.4 | −5.24 | −5.24 | −5.44 | −5.44 | −5.44 | −5.29 | −5.49 | −5.97 | −5.59 | −5.42 | −5.42 | −7.6 |
| ESOL Class | MS | MS | MS | MS | MS | MS | MS | MS | MS | MS | MS | MS | PS |
| GI absorption | High | High | High | High | High | High | High | High | High | High | High | High | Low |
| BBB permeant | Yes | Yes | Yes | Yes | Yes | Yes | Yes | Yes | No | Yes | No | No | No |
| P−gp substrate | Yes | Yes | Yes | Yes | Yes | Yes | Yes | Yes | Yes | Yes | No | No | No |
| CYP1A2 inhibitor | Yes | Yes | Yes | Yes | Yes | Yes | Yes | Yes | Yes | Yes | Yes | Yes | No |
| CYP2C19 inhibitor | Yes | Yes | Yes | Yes | Yes | Yes | Yes | Yes | Yes | Yes | Yes | Yes | Yes |
| CYP2C9 inhibitor | No | No | No | No | No | No | No | Yes | No | No | No | No | No |
| CYP2D6 inhibitor | Yes | Yes | Yes | Yes | Yes | Yes | Yes | Yes | No | Yes | Yes | Yes | No |
| CYP3A4 inhibitor | Yes | Yes | Yes | Yes | Yes | Yes | Yes | Yes | Yes | Yes | No | No | No |
| Lipinski #violations | 0 | 0 | 0 | 0 | 0 | 0 | 0 | 0 | 1 | 0 | 0 | 0 | 1 |

MW = Molecular weight; TPSA = Topological polar surface area; Log P = Logarithm of the partition coefficient; ESOL Log S = ESOL model logarithm of molar solubility in water; ESOL class = Solubility class in Log S scale; MS = Moderately soluble; PS = Poorly soluble; GI = Gastrointestinal; BBB = Blood–brain barrier; P-gp = Permeability glycoprotein; CYP = Cytochrome.

## 4. Conclusions

A series of 13 derivatives of 2,4,5-trisubstituted imidazole were synthesized and their structures were characterized and confirmed through a series of spectroscopic and spectrometric techniques. Their antioxidant activities were analyzed with DPPH radical-scavenging and ABTS radical cation scavenging assays. In DPPH, the most active compounds were **3** and **10** ($EC_{50}$ of 0.141 and 0.174 mg/mL, respectively), bearing a *p*-OH and *p*-dimethylamino substitution in their A ring; in ABTS, the most active compounds were again **10** and **3** with an $EC_{50}$ of 0.162 and 0.168 mg/mL, respectively. This suggests the important role of heteroatoms with a free pair of electrons and acid phenolic hydrogens, so future derivatives should maintain these characteristics for improved antioxidant activity.

In the enzymatic assays, though not as active as the controls, **1** showed the best activity in AChE inhibition with 25.8% of inhibition, followed by the nitro containing compounds **12** (22.4%) and **11** (21.2%). The most active XO inhibitor was **3**, with an $IC_{50}$ of 85.8 μg/mL and a *p*-OH substitution. Present results point out that aromatic and positively charged groups are important for AChE inhibition activity, as the literature suggests. For XO inhibition, an oxygen in the *para* position appears to improve triphenyl imidazole derivatives activities, though an unexpected result for compound **12** suggests that future derivatives with nitrogen in the *ortho* position should be further explored.

The antiproliferative activity was evaluated against six cell lines from different anatomic origins, and the synthesized compounds showed from moderate to very good activities. Amongst the most active compounds were **2** (*o*-OH), **10** (*p*-N(CH$_3$)$_2$), and **11** (*p*-NO$_2$), where the last was outstanding as it was the first or second most active against all of the evaluated cell lines. Further expansion of this family of derivatives could maintain a nitrogen in the *para* position of the A ring, as it appears this favors their antiproliferative activity, with additional structure modulations.

In the in silico analysis, the docking against the EGFR and HER2 receptors had the agreement of **11** being amongst the two better binding affinities results. The ADME predictions of the 13 synthesized compounds showed that they are overall suitable for oral administration, with **11** and **12** having better pharmacokinetics and drug-likeness properties, which combined with their in vitro results point them as good candidates for being lead compounds in further derivations in the search of new drugs, especially as AChE inhibitors or as antiproliferative agents.

**Supplementary Materials:** The following are available online at http://www.mdpi.com/2076-3417/10/8/2889/s1, Figure S1: 1H-NMR Compound 2, Figure S2: 1H-NMR extension Compound 2, Figure S3: 13C-NMR Compound 2, Figure S4: 1H-NMR Compound 4, Figure S5: 13C-NMR Compound 4, Figure S6: 1H-NMR Compound 7, Figure S7: 13C-NMR Compound 7, Figure S8: 1H-NMR Compound 8, Figure S9: 13C-NMR Compound 8, Figure S10: 1H-NMR Compound 9, Figure S11: 13C-NMR Compound 9, Figure S12: 1H-NMR Compound 11, Figure S13: 13C-NMR Compound 11.

**Author Contributions:** Conceptualization, I.C.-G.; Methodology, E.N.-I., J.M.P., R.S.-A., D.C., L.D.-R., and V.W.B.-C.; Software, A.E.-C. and V.G.-G.; Formal analysis, E.N.-I. and R.A.C.-S.; Investigation, I.C.-G. and R.D.-M.; Writing—original draft preparation, E.N.-I. and I.C.-G.; Writing—review and editing, A.E.-C., R.A.C.-S., R.D.-M., and L.D.-R.; Supervision, I.C.-G. and L.D.-R.; Funding acquisition, V.G.-G. and R.D.-M. All authors have read and agreed to the published version of the manuscript.

**Funding:** This research received no external funding.

**Acknowledgments:** We gratefully acknowledge the Facultad de Ciencias Químicas e Ingeniería, Universidad Autónoma de Baja California for the financing given for the realization of this project. E.N.-I. acknowledges CONACYT for the doctoral fellowship. J.M.P. thanks the Spanish Government for financial support through project PGC2018-094503-B-C22 (MCIU/AEI/FEDER, UE).

**Conflicts of Interest:** The authors declare no conflict of interest.

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
