# Peer review of "In Vitro and In Silico Screening of 2,4,5-Trisubstituted Imidazole Derivatives as Potential Xanthine Oxidase and Acetylcholinesterase Inhibitors, Antioxidant, and Antiproliferative Agents"

_applsci, doi:10.3390/app10082889_

Round 1

Reviewer 1 Report

The authors synthesized 2,4,5-trisubstituted imidazole derivatives, and investigated their antioxidant activity, and inhibitory effects on xanthine oxidase, cholinesterase, and proliferation of tumor cells. Synthesis and bioactivity of various 2,4,5-trisubstituted imidazole derivatives have been reported. The author should cite previous similar articles more and describe the novelty of their work. Overall, I think that this paper at present is not acceptable for publication. Only the critical points are written bellow.

Antioxidant activity of 2,4,5-trisubstituted imidazole derivatives including compound 2, 3 and 10 in this study has been reported in WJPR (2015) 4, 1321-1333. They should describe this point and differences from it.

In figure 1, does each compound decrease the activity significantly? The authors should show standard deviation or error on each bar.

Antiproliferative activity of 2,4,5-trisubstituted imidazole derivatives including compound 8 in this study has been reported in PLOS One (2016) DOI:10.1371/journal.pone.0153155. They should describe this point and differences from it.

The authors should describe the reason why they focused on EGFR and HER2 for the target molecules of 2,4,5-trisubstituted imidazole derivatives. Docking analysis without any experimental supporting data is less informative. The article described above suggested a 2,4,5-trisubstituted imidazole derivative abrogates PI3K/Akt/mTOR pathway. At least, the authors should investigate the antiproliferative activity of known EGFR and HER2 inhibitors on cells used in table 2.

Author Response

Thank you for the review you gave to our manuscript, and your observations of it. Please find attached a document containing our responses.

Reviewer 2 Report

The authors have submitted a manuscript in which they analyze the cytotoxic, antioxidant, AChE and XO activity of a series of 2,4,5-triphenyl imidazole derivatives, selecting the best candidates for new drug families thought in vitro and in silico analysis.

The topic of this paper is suitable for the scope of the Journal, but the novelty could be better explained. In particular I suggest adding a paragraph about the reason why these new 2,4,5-triphenyl imidazole derivatives could be useful in solving the problems mentioned in the Intruduction.

Moreover, more details and examples could be added in lines 50-54.

Paragraph 2.2, Lines 89-94: It would be better to use the past instead of the present tense. A better description of the synthesis would be appreciated (maybe it's enough to improve English)

Paragraph 3.1

Lines 288-291: Please, specify the reaction yields for the different derivatives

Paragraph 3.2

Lines 304-307: The discussion is not clear: revise English form

Lines 322-326: not clear: revise English form. Moreover, discuss more these results and try to explain the differences of results also in relation to the assay.

Paragraph 3.5

Line 399 different anatomic origins: please specify better

The Conclusions repeat the main numerical results. It would be preferable to underline better the main goals reached with this research, their perspectives and which future researches are needed to deepen and exploit your results. In this perspective, last sentence can be improved and elaborated.

Pay more attention to the layout of Figures S1–S13 of the NMR spectra and the derivative structures. Some figures are cut or overlap.

Other minor remarks:

Why are the references in yellow?

Line 188-189: use subscripts for A1 and A2

A huge revision of English grammar and form is necessary

e.g. Line 186, 203 and 238 “Expressed” instead of “expresses”

Line 194 reverse 980 µL of this solution

Line 207: reverse the English form (subject before the verb)

Line 209, 213 Reverse English form (subject before the verb)

Line 224 “in triplicate” instead of by

Line 228: what do you mean with “was supplemented with”?

Lines 255-257: revise English

Line 314: eliminate“,” between “imidazole” and “are”

Lines 318-321: split the paragraph into two sentences.

Line 331 “among” instead of “from”

Lines 337-340 revise English

Line 341 “The compounds with higher inhibition percentages were 11 and 12”

Line 345: revise English

Lines 355-357: revise English

Line 377: compound 10 “has” instead of “have”

Line 388: revise English

Line 390: it “has” the second lowest active instead of “is”

Lines 402-403: reverse imidazole was used

Line 418: “is in agreement” subject is missing

Line 432: were instead of are

Line 439: revise English

Line 440: The comparison between…showed

Line 467-468: revise English

Lines 487-489: revise English

Line 497: results instead of result

Line 515: as it can be seen

Author Response

(The authors gave the same response as above.)

Reviewer 3 Report

The authors presented interesting research results regarding the synthesis of 13 2,4,5-trisubstituted imidazole derivatives and screening of their activities, such as: antioxidant, cytotoxic, xanthine oxidase and cholinesterase inhibitors. I appreciate that tested activities are somehow related and it was also emphasized by the authors.

Following minor errors or some suggestions are presented below:

Page 1, line 35 and Page 8, line 338, and Page 16, line 562: In my opinion phrase “the better” should be replaced by “the best”. If you want to use “better” you should add, i.e. “13 had better activity than 11”

The red color of the font should be changed to black.

Page 2, line 91: It is Bruker not Brucker

In formulas (for both DPPH, ABTS and other methods, where there are percentage values) after the number 100 you should add the percent sign (%).

Page 6, line 249: "HBL-100, (breast)" - The comma separates it unnecessarily

When you are describing the results of the DPPH radical method, please also take into account the difference in activity for isomers - compounds 2 and 3.

Is Figure 3 is needed in the manuscript? The same observations as this: "Based on the GI50 range (Figure 3), the most active compounds of the series are 10 and 11" can be observed by analyzing Table 2.

The biggest complaint about this publication is the lack of description of the results of statistical analyzes. It is mentioned in Materials and Methods section that ANOVA was performed. Were post-hoc tests performed? There is no information on statistical analysis in figures and tables.

Author Response

Thank you for the review you gave to our manuscript, and your observations of it.

Reviewer 4 Report

Line 3 – there should be „oxidase” after “xanthine”

Abstract – I find it appropriate to add the chemical names of the compounds mentioned next to their numeral symbols the first time they appear.

I find it appropriate to change “cytotoxic” to “antiproliferative” in the whole manuscript, including the title, since the method used (sulforhodamine B staining) does not allow examining the membrane damage by the cytotoxic compounds.

Line 95 – it should be “Scientific”.

Line 225 – the sentence should be started with “The five…” rather than “5…”.

The Authors indicated in the Introduction that Alzheimer disease is caused among others by oxidative stress and cholinergic deficit. Moreover, the presented studies indicated that some of the tested compounds exhibited strong antioxidant and acetylcholinesterase inhibitor activity. My question is whether is there any compounds that exhibit both these activities at a satisfactory level. Discuss this in the context of its possible use as a medicine in Alzheimer treatment.  

Author Response

(The authors gave the same response as above.)

Reviewer 5 Report

The manuscript from a scientific point of view is very interesting and deserves to be published.

Author Response

Thank you for the review you gave to our manuscript, and your observations of it. Right now we are make a few adjustments to it that the other reviewers asked for, but we are confident that the manuscript will be published soon, so it will be readily available for the scientific community.

            Thanks again for your support.

Round 2

Reviewer 1 Report

The authors partially revised the manuscript and figures as the reviewers pointed out. However, it is still unclear what they found newly in this manuscript and there are many points that still need to be addressed.

Antioxidant activity:

Even though there are some differences in the method with the previous report, the authors still need to cite the article and describe why their data is better than previous one because this manuscript is not first report about the “antioxidant activity” of some compounds .  

Acetylcholinesterase inhibitory assay:

The authors should disucss more carefully based on which compounds showed “statistically significant” decrease. Also, comparison between the compounds with no significant difference is meaningless.

Antiproliferative activity and molecular docking:

An important thing is that molecular docking study itself cannot provide any direct evidence. Without any experimental supporting data, it is less informative. In the previous manuscript lines 471-474, they explained the importance of EGFR and HER2 in the tumor cells. However, they have not done why they could focus on EGFR and HER2 for “the target molecules” of the compounds in the many target candidate proteins.

Therefore, at least the authors need to show some correlation between antiproliferative activity and molecular docking to EGFR and HER2. For example, they should investigate the effects of the compounds together with known EGFR and HER2 inhibitors on growth of cells which express less HER2 and EGFR. Also, they need to evaluate the binding energies of imidazole and known EGFR and HER2 inhibitors as negative and positive control in Table 3.
